# Serum Adiponectin, a Novel Biomarker Correlates with Skin Thickness in Systemic Sclerosis

**DOI:** 10.3390/jpm12101737

**Published:** 2022-10-19

**Authors:** Giorgia Leodori, Chiara Pellicano, Valerio Basile, Amalia Colalillo, Luca Navarini, Antonietta Gigante, Francesca Gulli, Mariapaola Marino, Umberto Basile, Edoardo Rosato

**Affiliations:** 1Department of Translational and Precision Medicine, Sapienza University of Rome, 00189 Rome, Italy; 2Clinical Pathology Unit and Cancer Biobank, Department of Research and Advanced Technologies, IRCCS Regina Elena National Cancer Institute, 00144 Rome, Italy; 3Unit of Allergology, Clinical Immunology and Rheumatology, Campus Bio-Medico University of Rome, 00128 Rome, Italy; 4Clinical Biochemistry Laboratory, IRCCS “Bambino Gesù” Children’s Hospital, 00165 Rome, Italy; 5Department of Translational Medicine and Surgery, Section of General Pathology, “A. Gemelli” IRCCS, Catholic University of the Sacred Heart, 00168 Rome, Italy; 6Department of Laboratory and Infectious Disease Sciences, “A. Gemelli” IRCCS, Catholic University of the Sacred Heart, 00168 Rome, Italy

**Keywords:** adiponectin, biomarkers, skin thickness, systemic sclerosis

## Abstract

The aim was to evaluate the longitudinal association between basal serum adiponectin and repeated measurements of skin thickness during 12 months of follow-up in systemic sclerosis (SSc) patients. We enrolled SSc patients with disease duration > 2 years in a prospective observational study. Skin thickness was measured at baseline and after 12 months of follow-up with modified Rodnan skin score (mRSS). Baseline serum adiponectin was determined using a commercial ELISA kit. We enrolled 66 female SSc patients (median age 54 years, IQR 42–62 years). The median disease duration was 12 (IQR 8–16) years and median baseline serum adiponectin was 9.8 (IQR 5.6–15.6) mcg/mL. The median mRSS was 10 (IQR 6–18) at baseline and 12 (IQR 7–18) at follow-up. A significant correlation was observed between baseline serum adiponectin and disease duration (r = 0.264, *p* < 0.05), age (r = 0.515, *p* < 0.0001), baseline mRSS (r = −0.303, *p* < 0.05), and mRSS at follow-up (r = −0.322, *p* < 0.001). In multiple regression analysis, only mRSS at follow-up showed an inverse correlation with baseline serum adiponectin (β = −0.132, *p* < 0.01). The reduction in serum adiponectin levels is correlated with skin thickness.

## 1. Introduction

Autoimmune diseases are characterized by excessive immune activation mistakenly targeting tissue and organs. Chronic inflammation involved in these pathologies leads to destruction and dysfunction of the immune system. Understanding the role and mechanism of various actors involved in autoimmune diseases allows different approaches for management and drug treatment of patients.

Systemic sclerosis (SSc) is an autoimmune disease characterized by immune system activation, endothelial dysfunction and tissue fibrosis [1]. Progressive skin thickening is the hallmark of SSc, but the fibrotic process may also involve the lungs, gastrointestinal tract, cardiac tissue and kidneys [2].

Dermal fibrosis leads to attrition of dermal adipose deposits, a specialized tissue identified as dermal white adipose tissue (dWAT), involved in thermoregulation, wound healing, regulation of skin appendages and protection against infections [3]. Recent evidence has demonstrated that dWAT is an important source of myofibroblasts and that adipocyte-myofibroblasts transition could be one of the pivotal pathogenetic events in SSc [4]. Peroxisome proliferator-activated receptor gamma (PPAR-gamma) plays a key role in regulation of fibrosis, modulating fibroblasts activation and myofibroblasts differentiation. Several studies have demonstrated that PPAR-gamma agonists reduce tissue fibrosis in vivo [5,6,7,8]. Adiponectin, an adipokine with a powerful insulin sensitizing action, represents a serological marker of PPAR-gamma activity. Reduction of adiponectin is associated with insulin resistance, diabetes mellitus, metabolic syndrome, and cardiovascular diseases [9]. Adiponectin also displays anti-inflammatory functions and could have beneficial effects on cardiovascular and metabolic disorders including atherosclerosis and insulin resistance [10,11,12]. The biologic effects of adiponectin are mediated by two trans-membrane receptors AdipoR1 and AdipoR2 linked to adenosine monophosphate (AMP)-activated protein kinase [13]. Several studies demonstrated a reduction in circulating adiponectin levels in SSc patients. The reduction of serum level of adiponectin is associated with skin fibrosis [14,15,16,17,18,19]. In murine models of scleroderma, the adiponectin plays an antifibrotic role. In dermal fibrosis, loss of adipocytes and their differentiation into myofibroblasts are early events. In vivo, TGFβ, a profibrotic cytokine, induces transformation of adipose tissue into myofibroblasts. According to this evidence, adiponectin could represent an interesting therapeutic target of SSc fibrosis. In SSc, fibrosis of skin and internal organs was associated with loss of dWAT and reduced levels of circulating adiponectin [4,13,20,21,22].

The aim of this study is to assess the association of serum adiponectin with skin fibrosis at baseline and after 12 months of follow-up in SSc patients. 

## 2. Patients and Methods

### 2.1. Patients

Sixty-six consecutive SSc female patients (median age 54 years, IQR 42–62) fulfilling the American College of Rheumatology/European League Against Rheumatism Collaborative Criteria for SSc, were enrolled in this study from December 2018 to January 2019 [23]. The SSc patients were recruited from the Scleroderma Unit of Policlinico Umberto I-Sapienza University of Rome. Thirty-eight (57.6%) had diffuse cutaneous SSc (dcSSc) and twenty-eight (42.4%) had limited cutaneous SSc (lcSSc) according to Le Roy et al. [24]. Demographic and clinical features of SSc patients are shown in Table 1.

Exclusion criteria were disease duration < 2 years, active malignancies, acute and chronic kidney disease, cardiovascular or cerebrovascular events, a history of uncontrolled systemic hypertension, hyperlipidaemia, diabetes, coagulopathy, chemotherapy, or implantation of autologous adipose tissue-derived cells for the treatment of DUs. Pregnant or breastfeeding women were excluded. 

Therapies with potential impact on skin fibrosis (corticosteroids at an equivalent dose of prednisone ≥ 10 mg/day, immunosuppressant, biologic drugs) and therapies with potential impact on adiponectin levels or PPAR-gamma function (thiazolinidinediones, angiotensin converting enzyme inhibitor, angiotensin II receptors blockers, statins) up to 1 year before enrolment, at enrolment or during study follow-up were also exclusion criteria. 

The study was conducted according to the Declaration of Helsinki. The subjects’ written consent was obtained, and the study was approved by the ethics committee of Sapienza University of Rome (IRB 377).

### 2.2. Clinical Assessment

All SSc patients underwent clinical examination with assessment of the main clinical indexes at baseline and every six months for a follow-up period of 12 months. At baseline, we extracted data on autoantibodies and biochemistry from the patients’ medical records.

Skin involvement was evaluated by the modified Rodnan Skin Score (mRSS) [25]. The mRSS was assessed by only one assessor (AG) at baseline and follow-up. The mRSS assessor was blinded to treatment and clinical features of SSc patients. The coefficient of variation for measurement of mRSS by the same observer on different days was 1.5%. According to recent observational data from EUSTAR SSc cohort, the progression of skin fibrosis was defined as an increase in mRSS ≥ 5 points and ≥25% from baseline to the 12 months follow-up observation [26,27]. 

The disease activity index (DAI) [28] and disease severity scale (DSS) [29] were used to evaluate the activity and severity of disease, respectively.

Nailfold video-capillaroscopy (NVC) was performed with a video-capillaroscope (Pinnacle Studio Version 8) equipped with a 500× optical probe. The NVC of the second, third and fourth finger was performed by the same blinded operator in each patient. According to Cutolo et al., the NVC patterns were defined as “early”, “active” and “late” scleroderma patterns [30].

### 2.3. Laboratory Procedures

Serum Adiponectin levels were determined at baseline using a commercial ELISA kit (Human Adiponectin Quantikine ELISA Kit, R&D Systems, Minneapolis, MN, USA), according to the instructions provided by the manufacturer. All samples to be tested in duplicate blood samples were obtained from the study group after a 12 h fasting period and were centrifuged at 2000 rpm for 10 min and stored at −80 °C until analyses. Median serum adiponectin level was reported as median and IQR and was expressed as mcg/mL. The minimum detectable dose of human Adiponectin ranged from 0.079–0.891 ng/mL. In healthy controls, the mean value of serum level of adiponectin is 6.641 mcg/mL (range 0.865–21.424) according to the manufacturer’s indications. All determinations were performed by an operator blinded to treatment and clinical features of the handled sample. Each sample was tested twice to minimize eventual discrepancies, and all tests were performed in the same laboratory with the same instruments.

### 2.4. Statistical Analysis

All statistical analyses were performed using the software SPSS, version 25.0. The normal distribution of data was evaluated by the Shapiro–Wilk test. All results were expressed as median and IQR. Categorical variables were expressed as a number and percentage. The comparisons of continuous variables were performed using Student’s *t*-test or Mann–Whitney test, as appropriate. Categorical variables were compared using the Chi-square test or Fisher’s exact test, as appropriate. Correlations between continuous variables were evaluated using Pearson’s r coefficient or the Spearman rank order correlation coefficient. Multiple regression analysis (insert) was used to evaluate the correlation between serum level of adiponectin (dependent variable) and continuous independent variables. In multiple regression analysis, we insert only the continuous variables which showed a significant linear correlation with serum level of adiponectin level: disease duration and mRSS at the end of follow-up. *p*-values < 0.05 were considered significant.

## 3. Results

Demographic and clinical characteristics of SSc patients are shown in Table 1. The median duration of disease was 12 years (IQR 8–16 years). Forty-one (62.1%) patients were positive for anti-topoisomerase I (Scl70) antibodies and 22 (33.3%) patients had anti-centromere (ACA) antibodies. NVC pattern was early in seventeen (25.8%) patients, active in 23 (34.8%) patients and late in 26 (39.4%) patients. At baseline, the median values of mRSS, DAI and DSS were 10 (6–18), 2.0 (1–4.5) and 4.0 (3–7), respectively. The average BMI was 22.7 (IQR 19.4–26) kg/m^2^).

After a 12-months follow-up, the median value of mRSS was 12 (7–18). We observed an increase of five points of mRSS from the baseline in four (6%) SSc patients. At the follow up, DAI and DSS were 3 (1–5) and 6 (3–10), respectively. 

In SSc patients, median serum adiponectin level was 9.8 (5.6–15.6) mcg/mL. It was significantly reduced in dcSSc compared to lcSSc [8.2 (4.2–13.3) mcg/mL vs. 14 (9–17.6) mcg/mL, *p* < 0.01]. Median serum adiponectin level was lower, but not significant, in Scl70 positive patients than in ACA positive patients [8.2 (IQR 4.5–14.8) mcg/mL vs. 13.6 (IQR 8.5–17.3) mcg/mL, *p* > 0.05]. There was no significant difference (*p* > 0.05) in median serum adiponectin level in NVC patterns [early 9.8 (5.6–14.4) mcg/mL, active 12.0 (5.0–16) mcg/mL and late 9.3 (5.7–15.2) mcg/mL]. 

No significant correlation (*p* > 0.05) exists between baseline serum adiponectin level and BMI, DAI and DSS. We found a significant correlation between baseline serum adiponectin and disease duration (r = 0.264, *p* < 0.05), age (r = 0.515, *p* < 0.0001), baseline mRSS (r = −0.303, *p* < 0.05, Figure 1), mRSS at the end of follow-up (r = −0.322, *p* < 0.001, Figure 2).

In multiple regression analysis adjusted for age, we insert only the continuous variables which showed a significant linear correlation with serum level of adiponectin level: disease duration and mRSS at the end of follow-up. In multiple regression analysis, mRSS at 12 months after follow-up shows a correlation with baseline serum adiponectin (β = −0.132, *p* < 0.01).

## 4. Discussion

This study demonstrated an inverse correlation of baseline serum adiponectin with skin fibrosis, assessed by mRSS, at baseline and after 12 months of follow-up. 

At baseline, we found decreased serum adiponectin in dcSSc patients compared to lcSSc. Moreover, adiponectin levels are lower in patients with Scl70 antibodies than in SSc patients with ACA antibodies. Lakota et al. demonstrated that patients with dcSSc had reduced serum adiponectin levels and mRSS showed an inverse correlation with serum adiponectin [15]. In a selected population of 36 dcSSc patients, Winsz-Szczotka et al. demonstrated that adiponectin was significantly lower in dcSSc patients compared to healthy controls. Adiponectin correlated significantly with leptin, total lipid peroxide, C-reactive protein, erythrocyte sedimentation rate and duration of disease. The authors conclude that adipose tissue may play a complex role in the development of dcSSc, affecting both the metabolic state of the organism, as well as free radical-induced connective tissue degradation [31]. Other authors associated progress of the disease with a reduction in the amount of adipose tissue and leptin level in SSc patients [32].

The anti-fibrotic potential of adiponectin is evident in mice models. Indeed, adiponectin knocked-out mice treated with bleomycin develop exaggerated skin fibrosis, while skin fibrosis is attenuated in adiponectin-overexpressing transgenic mice [21]. Recent evidence clarified that adiponectin mediates the anti-fibrotic action [13,21]. In recent years it has been clarified the role of dWAT in the pathogenesis of skin fibrosis, a hallmark of SSc. Peroxisome proliferator-activated receptor gamma (PPAR-gamma), a pleiotropic nuclear receptor, plays a pivotal role in regulation of fibrosis, modulating fibroblasts activation and myofibroblasts differentiation. Adiponectin is a specific index of PPAR-gamma activity and displays also anti-inflammatory functions and its levels are reduced in several systemic rheumatic diseases. The biologic effects of adiponectin are mediated by two trans-membrane receptors AdipoR1 and AdipoR2 linked to adenosine monophosphate (AMP)-activated protein kinases. Recent data from murine models of scleroderma, deepening the myofibroblasts differentiation and the role of PPAR-gamma, uncover the implication of adiponectin in the process of fibrosis. According to this evidence, adiponectin is involved in inhibition of skin fibrosis in scleroderma and represent an interesting therapeutic target [4,21]. According to these findings, adiponectin could be a potential therapeutic approach to control dWAT attrition and myofibroblast development [21]. In our study, a positive correlation exists between disease duration and basal adiponectin levels. That is rather discrepant, since basal adiponectin level is reduced with increasing fibrosis. The early SSc disease is characterized by the maximum progression rate of skin thickness, in a temporal window not clearly defined but considered around the first 18 months from thickness onset. 

The possible explanation is that skin fibrosis can be independent from disease duration after early SSc disease characterized by the maximum progression rate of skin thickness. To avoid the confounding impact of different disease phases on data analysis, requiring stratification of patients according to disease duration strata, we decided to enrol patients with disease duration > 2 years. The median disease duration of SSc sample enrolled in this study was about 12 years. In a recent study from the EUSTAR group on patients with dcSSc, short disease duration, low baseline mRSS and joint synovitis were identified as independent predictors of progressive skin fibrosis within 1 year [26]. 

The focus of precision medicine is to identify effective therapeutic approaches for patients based on various factors. Major topics of our research lines include immunological investigation on novel diagnostic biomarkers. In recent studies, we demonstrated that complement, free light chains and B Cell phenotype are new markers of disease [33,34,35].

This study has several limitations, such as a small sample size, pre-selected population, absence of skin biopsy at baseline and at follow-up, and especially the absence of serum level adiponectin assessment at follow-up. 

In conclusion, the serum levels of adiponectin were reduced in dcSSc patients. The inverse correlation between serum level of adiponectin and mRSS was observed at baseline and after 12 months of follow-up.

## Figures and Tables

**Figure 1 jpm-12-01737-f001:**
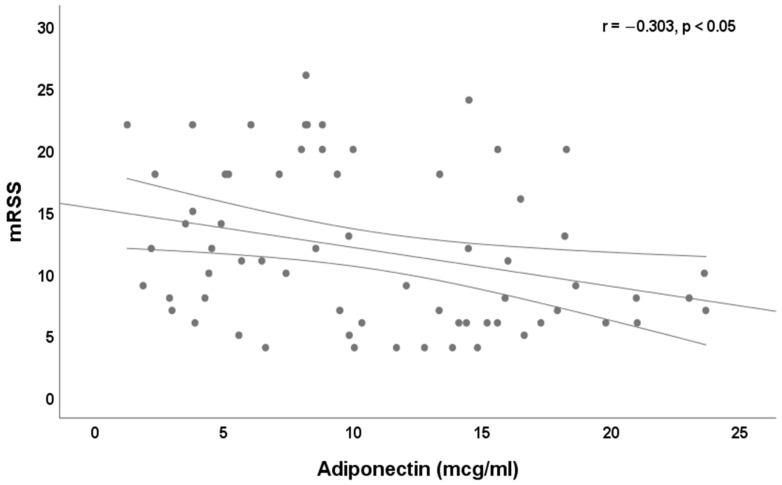
Correlation between baseline serum level of adiponectin (mcg/mL) and baseline mRSS (r = −0.303, *p* < 0.05).

**Figure 2 jpm-12-01737-f002:**
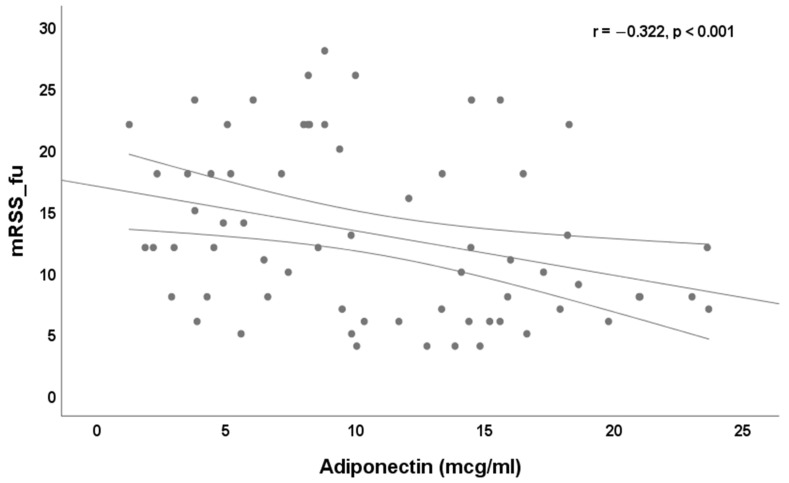
Correlation between baseline serum level of adiponectin (mcg/mL) and mRSS at the end of follow-up (r = −0.322, *p* < 0.001).

**Table 1 jpm-12-01737-t001:** Demographic and clinical correlates of SSc patients.

Females, *n* (%)	66 (100)
Age, years, median and IQR	54 (42–62)
dcSSc, *n* (%)	38 (58)
SSc specific autoantibodies	
anti-topoisomerase I, *n* (%)	41 (62.1)
anti-centromere, *n* (%)	22 (33.3)
none, *n* (%)	3 (4.5)
Nailfold capillaroscopic pattern	
early, *n* (%)	17 (25.8)
active, *n* (%)	23 (34.8)
late, *n* (%)	26 (39.4)
Adiponectin, ng/mL, median and IQR	9.8 (5.6–15.6)
mRSS, median and IQR	10 (6–18)
mRSS_fu, median and IQR	12 (7–18)
DAI, median and IQR	2 (1–4.5)
DAI_fu, median and IQR	3 (1–5)
DSS, median and IQR	4 (3–7)
DSS_fu, median and IQR	6 (3–10)
BMI, kg/m^2^, mean ± SD	22.7 (19.4–26)

## Data Availability

The data presented in this study are a vailable on request from the corresponding authors.

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
