# Peer review of "Serum Adiponectin, a Novel Biomarker Correlates with Skin Thickness in Systemic Sclerosis"

_jpm, 2022, doi:10.3390/jpm12101737_

Round 1

Reviewer 1 Report

The article is of interest to clinicians and researchers.

Introduction requires a better description of the potential role of adiponectin in Scleroderma. 

Results are confusing. Can you explain the discrepancy between the low correlation coefficient and significant p values?

Author Response

Reviewer 1

The article is of interest to clinicians and researchers.

Reply: We thank the reviewer for her/his comment

Introduction requires a better description of the potential role of adiponectin in Scleroderma. 

Reply: According to reviewer we modified the introduction and we added a better description of the potential role of adiponectin in SSc.

Results are confusing. Can you explain the discrepancy between the low correlation coefficient and significant p values?

Reply: we modified the result reporting all data of follow-up in one paragraph. In our results we observe a weak correlation for r<0,3 (p<0.05), middle correlation for r=0.3-0,35 and strong correlation for r>0.35

Reviewer 2 Report

1. study design, study site and study population should be described well

2.  method section should be well explained for reproducibility

3. conclusion is conspicuously missing

4. few incomplete sentences/ spelling mistakes

Autoimmune diseases are characterized by excessive immune activation mistank-
enly targets tissue and organ.

Chronic inflammation involves in these pathologies de-struction and dysfunction of immune system.

Author Response

  1. study design, study site and study population should be described well

Reply: according to reviewer we added the site of the study. The others information were already present in the paragraph.

  1. method section should be well explained for reproducibility

Reply: The section was divided in 4 paragraphs to explain better methods of study

  1. conclusion is conspicuously missing

Reply: we thank the reviewer for this important suggestion. The conclusions were rewritten.

  1. few incomplete sentences/ spelling mistakes

Reply: we thank the reviewer for his/her comment. We corrected the errors.

Autoimmune diseases are characterized by excessive immune activation mistank-

enly targets tissue and organ.

Chronic inflammation involves in these pathologies de-struction and dysfunction of immune system.

Reviewer 3 Report

An interesting study of the predicting role of serum adiponectin level in development of cutaneous changes in patients with systemic sclerosis. I have no major comments. I suggest to the Authors to consider the following minor changes in the manuscript:

-          Adiponectin is produced in majority by adipose tissue. It is possible that progress of the disease is associated with reduction of the amount of adipose tissue in the patients. It is in agreement with reduced leptin level in patients with systemic sclerosis (there is a mention in the manuscript but without reference, e.g. Kotulska A. et al.: A decreased serum leptin level in patients with systemic sclerosis. Clin Rheumatol 2001; 20: 300-302). I suggest to add such comment in the discussion.

-          The correct name of the disease is: systemic sclerosis but not Systemic Sclerosis (without capitalization of the first letters).

Author Response

Reviewer 3

An interesting study of the predicting role of serum adiponectin level in development of cutaneous changes in patients with systemic sclerosis. I have no major comments. I suggest to the Authors to consider the following minor changes in the manuscript:

Reply: we thank the reviewer for his/her positive evaluation.

-          Adiponectin is produced in majority by adipose tissue. It is possible that progress of the disease is associated with reduction of the amount of adipose tissue in the patients. It is in agreement with reduced leptin level in patients with systemic sclerosis (there is a mention in the manuscript but without reference, e.g. Kotulska A. et al.: A decreased serum leptin level in patients with systemic sclerosis. Clin Rheumatol 2001; 20: 300-302). I suggest to add such comment in the discussion.

Reply: We added the sentence in discussion with reference

-          The correct name of the disease is: systemic sclerosis but not Systemic Sclerosis (without capitalization of the first letters).

Reply: we corrected the error